# WIN55212-2 Modulates Intracellular Calcium via CB_1_ Receptor-Dependent and Independent Mechanisms in Neuroblastoma Cells

**DOI:** 10.3390/cells11192947

**Published:** 2022-09-21

**Authors:** Victor M. Pulgar, Allyn C. Howlett, Khalil Eldeeb

**Affiliations:** 1Department of Pharmaceutical and Clinical Sciences, College of Pharmacy and Health Sciences, Campbell University, Buies Creek, NC 27506, USA; 2Biomedical Research and Infrastructure Center, Winston-Salem State University, Winston-Salem, NC 27101, USA; 3Department of Obstetrics & Gynecology, Wake Forest School of Medicine, Winston-Salem, NC 27157, USA; 4Department of Physiology & Pharmacology, Wake Forest School of Medicine, Winston-Salem, NC 27157, USA; 5Jerry M. Wallace School of Osteopathic Medicine, Campbell University, Buies Creek, NC 27506, USA; 6AL Azhar Faculty of Medicine, New Damietta 34518, Egypt

**Keywords:** Ca^2+^ mobilization, receptor crosstalk, cannabimimetic aminoalkylindoles

## Abstract

The CB_1_ cannabinoid receptor (CB_1_R) and extracellular calcium (eCa^2+^)-stimulated Calcium Sensing receptor (CaSR) can exert cellular signaling by modulating levels of intracellular calcium ([Ca^2+^]_i_). We investigated the mechanisms involved in the ([Ca^2+^]_i_) increase in N18TG2 neuroblastoma cells, which endogenously express both receptors. Changes in [Ca^2+^]_i_ were measured in cells exposed to 0.25 or 2.5 mM eCa^2+^ by a ratiometric method (Fura-2 fluorescence) and expressed as the difference between baseline and peak responses (ΔF_340/380_). The increased ([Ca^2+^]_i_) in cells exposed to 2.5 mM eCa^2+^ was blocked by the CaSR antagonist, NPS2143, this inhibition was abrogated upon stimulation with WIN55212-2. WIN55212-2 increased [Ca^2+^]_i_ at 0.25 and 2.5 mM eCa^2+^ by 700% and 350%, respectively, but this increase was not replicated by CP55940 or methyl-anandamide. The store-operated calcium entry (SOCE) blocker, MRS1845, attenuated the WIN55212-2-stimulated increase in [Ca^2+^]_i_ at both levels of eCa^2+^. Simultaneous perfusion with the CB_1_ antagonist, SR141716 or NPS2143 decreased the response to WIN55212-2 at 0.25 mM but not 2.5 mM eCa^2+^. Co-perfusion with the non-CB_1_/CB_2_ antagonist O-1918 attenuated the WIN55212-2-stimulated [Ca^2+^]_i_ increase at both eCa^2+^ levels. These results are consistent with WIN55212-2-mediated intracellular Ca^2+^ mobilization from store-operated calcium channel-filled sources that could occur via either the CB_1_R or an O-1918-sensitive non-CB_1_R in coordination with the CaSR. Intracellular pathway crosstalk or signaling protein complexes may explain the observed effects.

## 1. Introduction

Phytocannabinoids, endocannabinoids, synthetic cannabinoids, and aminoalkylindole agonists regulate neuronal activity by activating the CB_1_ receptor (CB_1_R) to signal via Gi/o and other G proteins, but little is known about modulating intracellular calcium concentration ([Ca^2+^]_i_). Although initial studies could not detect Ca^2+^ mobilization in cultured cell models, evidence indicates that the effects of cannabinoid receptors on [Ca^2+^]_i_ depend on the agonist and the cell type tested. In CHO cells, the synthetic cannabinoid HU210 and its non-CB_1_-binding isomer HU211 (10 μM) both induced only a non-receptor-mediated increase in [Ca^2+^]_i_ in untransfected, CB_1_R-expressing, or CB_2_R-expressing cells, using a Fura-2 method that readily detected muscarinic receptor-mediated Ca^2+^ mobilization [1,2]. In the murine neuroblastoma cell line N18TG2 endogenously expressing CB_1_R (but not CB_2_R), treatment with the non-selective CB_1/2_R agonists 2-arachidonoylglycerol (2-AG), CP55940, or WIN55212-2 (up to 1 μM) failed to evoke Ca^2+^ mobilization using Fluo-4 fluorescence able to detect a bradykinin-mediated response [3]. The Sugiura laboratory investigated Ca^2+^-deprived suspensions of HL60 monocytic cells endogenously expressing CB_2_R or NG108-15 neuro-glioma hybrid cells endogenously expressing CB_1_R. In this setting, Ca^2+^ mobilization was stimulated by 1 mM eCa^2+^ followed by 2-AG, CP55940, or WIN55212-2 (up to 10 μM) and detected with Fura-2 [4,5]. Subsequent studies showed that the aminoalkylindole WIN55212-2, but not Δ^9^-tetrahydrocannabinol (Δ^9^-THC), HU210, CP55940, 2-AG, or methanandamide, increased [Ca^2+^]_i_ in HEK293 cells exogenously expressing CB_1_R [6]. The WIN55212-2-stimulated Ca^2+^ mobilization occurred in a CB_1_-dependent manner requiring Gα_q_ activation and release of Ca^2+^ from thapsigargin-sensitive endoplasmic reticulum (ER) stores [6]. In addition, GPR55 and GPR18 receptors have been shown to modulate [Ca^2+^]_i_ in neurons and other cell types in response to lipid mediators, including atypical cannabinoids [7].

Mounting evidence supports intracellular Ca^2+^ as an important second messenger for excitable and non-excitable cells, with the inositol trisphosphate (IP_3_)/Calcium signaling pathway playing a vital role in linking extracellular signals to [Ca^2+^]_i_ [8]. Thus, one of the receptor systems involved, the G protein-coupled calcium sensing receptor (CaSR) detects extracellular Ca^2+^ (eCa^2+^) concentration, linking it to intracellular signaling affecting cell function [9]. The CaSR can couple to more than one type of Gα subunit and influence the properties of Gβγ signaling [10]. CaSR actions have been reported to act through Gαi, Gα_q_, and Gβγ, with activation of phospholipase C, production of IP_3_ through Gα_q_, and Ca^2+^ release from the ER, being one of the major effects of CaSR activation [11]. In N18TG2 neuronal cells, stimulation of CaSR with the positive allosteric modulator calindol increased [Ca^2+^]_i_ in a response dependent on Gα_i/o_ and modulated by Gα_q_ [12]. Modulating [Ca^2+^]_i_ also seems dependent on PKC activity and localization [13]. It appears that the CaSR intracellular pathways activated by eCa^2+^ proceed via Gα_s_ and Gα_q_, whereas activation by calcimimetics occurs via Gα_i_ [9].

In the present study, we aimed to explore neuronal mechanisms involved in WIN55212-2-mediated Ca^2+^ mobilization as observed by Lauckner and colleagues [6], with a focus on the extracellular [Ca^2+^] influence associated with the Sugiura procedure [4,5]. We were particularly interested in the cannabinoid receptors mediating the WIN55212-2-dependent responses in [Ca^2+^]_i_ and the role of CaSR activation on those responses. [Ca^2+^]_i_ regulation has relevant physiological significance, for example, in muscle [14] and brain tissue [15], where a role for CB_1_Rs has been demonstrated. Since the CaSR monitors the extracellular Ca^2+^ environment, our studies were performed in the N18TG2 neuroblastoma cell model that endogenously expresses both CB_1_R and CaSR.

## 2. Materials and Methods

### 2.1. Cells

Mouse N18TG2 neuroblastoma cells were cultured as described [16], maintained in complete media containing Dulbecco’s Modified Eagle’s Medium (DMEM): Ham’s F-12 (1:1) supplemented with penicillin (100 U/mL) and streptomycin (100 μg/mL) and 10% heat-inactivated bovine serum. Cells were grown in 75-cm^2^ flasks at 37 °C in a humidified atmosphere (5% CO_2_), harvested at sub-confluency, and transferred to 12 mm glass coverslips (Fisher Scientific Co., Waltham, MA, USA). At 50–75% confluence, cells were loaded for 15 min with Fura-2 (5 µM) in Krebs–Henseleit Buffer (KHB) containing (in mM) NaCl 118, KCl 4.47, NaHCO_3_ 25, KH_2_PO_4_ 1.2, MgSO_4_ 1.2, CaCl_2_·2H_2_O 0.25, glucose 5.5. Cells were incubated in two different extracellular [Ca^2+^] (eCa^2+^) and the responses in intracellular [Ca^2+^] ([Ca^2+^]_i_) were measured.

### 2.2. Imaging

Coverslips were transferred to an imagining chamber on an inverted Olympus BBX51WI microscope equipped with a 40× objective, a xenon arc lamp (Sutter Instruments, Novato, CA, USA), and a manual stage, and a cooled charge-couple device (CCD) camera (Hamamatsu Orka II). For ratiometric imaging, the microscope was computer-controlled by HCImage software (Hamamatsu Corporation, Middlesex, NJ, USA). Cells on the field were manually marked for analysis, and F_340_ and F_380_ were measured for one min. KHB containing 0.25 mM Ca^2+^ (Low eCa^2+^) was passed through the imaging chamber for 5 min after which eCa^2+^ was changed to 2.5 mM (High eCa^2+^) using a perfusion valve control system (VC-6, Six Channel Perfusion Valve Control Systems, Warner Instruments, Holliston, MA, USA) and cells perfused for additional 10 min. This procedure was repeated in the presence of cannabinoid receptor agonists (WIN55212-2, CP55940, or methanandamide) in KHB containing low or high eCa^2+^. Cells were pre-incubated for 15 min for the treatments with antagonists, and the corresponding antagonists were added to the low and high eCa^2+^ solutions. Only one treatment was carried out on each coverslip used.

### 2.3. Drugs

The ratiometric fluorescent dye Fura-2 was purchased from Molecular Probes (Eugene, OR, USA), dissolved in dimethylsulfoxide at 1 mM, and further diluted in KHB containing 0.25 mM Ca^2+^ to a working concentration of 5 μM as described [17]. The aminoalkylindole agonist of CB_1/2_ cannabinoid receptors WIN55212-2 (5 μM) [6], the prototype bicyclic non-selective CB_1/2_R agonist CP55940 (5 μM) [6], and the stable chiral analog of anandamide, a CB_1_R partial agonist, methanandamide (5 μM) [6], were from Cayman Chemical Co, (Ann Arbor, MI, USA). Receptor antagonists used include the blocker of CaSR, NPS2314 (3 μM) [18], the selective store-operated calcium (SOC) channel inhibitor N-propargyl-nitrendipine (MRS1845, 10 μM) [19], the CB_1_R antagonist SR141716 (1 μM) [6], and the non CB_1_/CB_2_ receptor blocker O-1918 (10 μM) [20], all from Cayman Chemical Co, (Ann Arbor, MI, USA). The cannabinoid compounds were stored at −20 °C as 10 mM stock solutions in ethanol. Immediately before use, an aliquot of drug stocks was air-dried and re-suspended in 0.25 mM Ca^2+^ KHB. All other chemical reagents were from Sigma-Aldrich Chemical Co. (St. Louis, MO, USA).

### 2.4. Data Analysis

Average changes in F_340_ and F_380_ were recorded continuously, and [Ca^2+^]_i_ responses were determined using the ratiometric method (ratio between F_340_ and F_380_) [17]. Imaging measurements were repeated 5 to 7 times with a total of 200 to 300 cells analyzed per each condition, and the background was subtracted automatically. Results were expressed as the difference between baseline and peak response (ΔF_340/380_), with data expressed as mean ±SEM (*n* = 5–7). Statistical analyses were performed by One-Way Analysis of Variance (ANOVA) and Newman-Keuls multiple comparisons test for data obtained in 0.25 or 2.5 mM Ca^2+^ using GraphPad Prism v6 (GraphPad Software Inc, La Jolla, CA, USA). A *p* < 0.05 was accepted as an indication of statistical significance.

## 3. Results

### 3.1. WIN55212-2 Increased [Ca^2+^]_i_ in N18TG2 Cells at Both Low eCa^2+^ and during a High-eCa^2+^ Stimulus

After resting at 0.25 mM extracellular Ca^2+^, perfusion of N18TG2 cells with 0.25 mM eCa^2+^ increased [Ca^2+^]_i_ transiently by 11%. When eCa^2+^ was changed to 2.5 mM, [Ca^2+^]_i_ increased by 304% over basal (ΔF_340/380_ 0.11 ± 0.04 vs. 0.334 ± 0.06 *p* < 0.05, Figure 1A,B). We tested the Ca^2+^ mobilization response to the non-selective CB_1/2_R aminoalkylindole agonist WIN55212-2 at concentrations that have previously been demonstrated to stimulate Ca^2+^ mobilization in HEK293 cells [6]. In the presence of WIN55212-2 (5 μM), [Ca^2+^]_i_ increased by 700% over basal in 0.25 mM eCa^2+^ (ΔF_340/380_ 0.11 ± 0.04 vs. 0.84 ± 0.12 *p* < 0.05) and by 350% over basal in 2.5 mM eCa^2+^ (ΔF_340/380_ 0.334 ± 0.06 vs. 1.29 ± 0.13 *p* < 0.05, Figure 1B–D).

### 3.2. CaSR Mediates Increases in eCa^2+^-Induced [Ca^2+^]_i_ in N18TG2 Cells

NPS2143 is a CaSR negative allosteric modulator that binds to the 7-transmembrane domain of the CaSR to inhibit the Ca^2+^ mobilization signaling pathway [21,22]. We used this calcilytic agent at concentrations previously shown to block increases in [Ca^2+^]_i_ promoted by activation of the Ca^2+^ receptor in HEK293 cells expressing the human Ca^2+^ receptor [18]. High eCa^2+^-induced [Ca^2+^]_i_ increase was effectively antagonized with NPS2143 (3 μM) (ΔF_340/380_ 0.36 ± 0.06 vs. 0.16 ± 0.02, *p* < 0.05, 56% of reduction, Figure 2A), demonstrating the functional activity of the CaSR evident at supra-physiological eCa^2+^. The WIN55212-2-induced elevations in [Ca^2+^]_i_ in low eCa^2+^ conditions were also attenuated by simultaneous perfusion with the NPS2143 (3 μM) (ΔF_340/380_ 0.84 ± 0.12 vs. 0.54 ± 0.05, *p* < 0.05, 36% of reduction, Figure 2B). Interestingly, in the presence of high eCa^2+^, the WIN55212-2-induced [Ca^2+^]_i_ increase was not inhibited by NPS2143 (ΔF_340/380_ 1.29 ± 0.13 vs. 1.25 ± 0.14, *p* > 0.05). These findings might suggest that the WIN55212-2 can influence [Ca^2+^]_i_ under “basal” CaSR conditions, but the WIN55212-2 stimulus was not influenced by NPS2143-inhibited CaSR. An alternative interpretation is that WIN55212-2 provided a mechanism to protect the CaSR from inhibition by the negative allosteric modulator.

### 3.3. Aminoalkylindole-Specific Potentiation of the eCa^2+^-Mediated Increase in [Ca^2+^]_i_

To check the selectivity of WIN55212-2-induced increase in [Ca^2+^]_i_ in neuronal cells, the non-classical cannabinoid full agonist, CP55940, and endocannabinoid partial agonist, methanandamide (Me-AEA) were used at concentrations previously shown to inhibit cAMP accumulation [23]. Both compounds at 5 μM failed to significantly increase [Ca^2+^]_i_ relative to basal values at either 0.25 mM (ΔF_340/380_ CP 0.29 ± 0.08, Me-AEA 0.05 ± 0.01, *p* > 0.05) or at 2.5 mM eCa^2+^ (ΔF_340/380_ CP 0.44 ± 0.03, Me-AEA 0.75 ± 0.17, *p* > 0.05, Figure 3). The responses to either CP55940 or Me-AEA on [Ca^2+^]_i_ at both levels of eCa^2+^ were significantly lower than the response to WIN55212-2 (*p* < 0.05). These findings support the WIN55212-2 selectivity, implicating a non-CB_1_ and non-CB_2_ mechanism for this response.

### 3.4. WIN55212-2-Stimulated Increases in [Ca^2+^]_i_ Require Operational Store Operated Calcium Entry (SOCE)

Ca^2+^ mobilization by GPCR-mediated production of inositol triphosphate (IP_3_) promotes Ca^2+^ release from ER stores, which requires continuous repletion via store operated Ca^2+^ entry (SOCE) mechanisms [24,25]. The most effective SOCE mechanism is based upon the ER [Ca^2+^] sensor stromal interacting molecule (STIM) and its association and activation of Ca^2+^ release-activated Ca^2+^ channels (CRAC) comprised of Orai1, Orai2, and Orai3 proteins. Cells also utilize non-selective cation channels as store-operated channels (SOCs), comprised of both Orai1 and transient receptor potential canonical channel 1 (TRPC1) channel subunits. Current reviews describe these processes in detail [25,26,27,28].

To evaluate the role of SOCE in WIN55212-2-stimulated increases in [Ca^2+^]_i_, we employed the Orai1 inhibitor N-propargyl-nitrendipine (MRS1845), which has a reported IC_50_ = 1.7 μM to block capacitative Ca^2+^ influx in HL60 cells [19], and also inhibits the ER Ca^2+^ replacement via TRPC1 at higher concentrations [29]. The WIN55212-2-stimulated increases in [Ca^2+^]_i_, in both eCa^2+^ conditions (0.25 mM or 2.5 mM Ca^2+^) were attenuated by incubation with MRS1845 (10 μM) (ΔF_340/380_ MRS 0.33 ± 0.07, *n* = 4, 61% reduction at 0.25 mM; ΔF_340/380_ MRS 0.7 ± 0.14, *n* = 7, 46% reduction at 2.5 mM, Figure 4, *p* < 0.05). These results are consistent with a requirement for continuous refilling of the intracellular Ca^2+^ stores in the ER as the source of the mobilized Ca^2+^.

### 3.5. WIN55212-2-Dependent Increases in [Ca^2+^]_i_ Are Mediated by either CB_1_R or a nonCB_1_/CB_2_ Receptor as a Function of the eCa^2+^ Stimulus

The N18TG2 neuronal cell expresses CB_1_R but fails to express CB_2_R [30,31,32], and thus, cellular signaling via cannabinoid receptors is expected to be inhibited by a CB_1_R competitive antagonist/inverse agonist such as SR141716 in this model [33]. Several non-CB_1_, non-CB_2_ GPCRs have been promoted as “Cannabinoid Related” receptors based on their ability to be orthosterically stimulated/inhibited or allosterically modified by phytocannabinoid or endocannabinoid-like compounds (see [34,35,36] for review). The cannabinoid related GPCRs GPR18 and GPR55 both signal through Ca^2+^ mobilization, and both interact with endocannabinoid-like N-arachidonoylglycine and N-arachidonoylserine, phytocannabinoid CBD, and CBD analogs abnormal-cannabidiol (abn-CBD), O-1602 and O-1918 [37]. For this reason, we chose to test O-1918 for its potential as an inhibitor of Ca^2+^ mobilization in these studies, and we selected a concentration of O-1918 (10 μM) that has been shown to block cannabinoid-dependent effects that are independent of CB_1_R or CB_2_R [20]. The WIN55212-2-induced increase in [Ca^2+^]_i_ in 0.25 mM eCa^2+^ was partially blocked by simultaneous perfusion with the CB_1_R antagonist SR141716 (1 μM) (ΔF_340/380_ SR 0.45 ± 0.07, *n* = 6, *p* < 0.05, 46% reduction). Under conditions of 2.5 mM eCa^2+^, SR141716 does not affect the WIN55212-2-dependent increase in [Ca^2+^]_i_ (ΔF_340/380_ SR 1.57 ± 0.11, *n* = 6, *p* > 0.05). Simultaneous perfusion with the nonCB_1_/CB_2_ receptor antagonist O-1918 (10 μM), attenuated WIN55212-2-promoted increases in [Ca^2+^]_i_ at both eCa^2+^ levels (ΔF_340/380_ O-1918 0.28 ± 0.07, *n* = 5, *p* < 0.05, 67% reduction at 0.25 mM; ΔF_340/380_ O-1918 0.72 ± 0.05, *n* = 5, *p* < 0.05, 44% reduction at 2.5 mM, Figure 5B). These findings implicate the role of the CB_1_R in WIN55212-2-promoted [Ca^2+^]_i_ increases in the absence of a CaSR stimulus. On the other hand, a prominent influence of a nonCB_1_/CB_2_ stimulus appears under conditions of activation of CaSR by supra-physiological eCa^2+^.

## 4. Discussion

The aminoalkylindole WIN55212-2 can modulate [Ca^2+^]_i_ in neuroblastoma cells via at least two mechanisms. At low eCa^2+^, the WIN55212-2 induced potentiation of [Ca^2+^]_i_ partially depends on CB_1_R defined by its sensitivity to inhibition by SR141716. At supra-physiologic eCa^2+^, which activates the CaSR, the effect of WIN55212-2 is CB_1_R-independent. At both eCa^2+^ levels, the release of Ca^2+^ is from intracellular stores filled by store-operated Ca^2+^ channels.

We observed that the effect of WIN55212-2 on [Ca^2+^]_i_ depends on the eCa^2+^ level. At low eCa^2+^, WIN55212-2 increases [Ca^2+^]_i_ acting via CB_1_R and nonCB_1_/CB_2_ receptors, probably acting on different intracellular transduction pathways. Actions of CB_1_R through pertussis toxin-sensitive G_i/o_ proteins leading to inhibition of cAMP production were first demonstrated in N18TG2 neuroblastoma cells [38]. CB_1_R acting through Gα_i_ could serve a modulatory role, as has also been proposed for Gα_q_ signaling, in mediating [Ca^2+^]_i_ increases in these cells [12]. The requirement for Gα_q_ on increasing [Ca^2+^]_i_ after CB_1_R activation was demonstrated in HEK293 cells and hippocampal neurons [6].

Our results of an eCa^2+^-dependent elevation in [Ca^2+^]_i_ confirm a role for CaSR in the modulation of [Ca^2+^]_i_ in N18TG2 neuroblastoma cells, as previously demonstrated [12]. The effects of WIN55212-2 we report at low and high eCa^2+^ were attenuated by SOCE blockade, suggesting that WIN55212-2 promotes the release of Ca^2+^ from intracellular stores. The main transduction pathway associated with this particular increase in [Ca^2+^]_i_ is described as dependent on Gα_q_, phospholipase C (PLC) activation, synthesis of diacylglycerol (DAG), and inositol triphosphate (IP_3_) with further activation of ER IP_3_ receptors promoting Ca^2+^ release [11]. PLC can be activated by either Gα_q_ or G_i/o_ βγ subunits, with these two effectors interacting with distinct regions of PLCs; Gα_q_ binds to the C-terminal and Gβγ binds to the catalytic domain [39]. Gα_q_ and G_i/o_ βγ can cooperate synergistically, increasing [Ca^2+^]_i_ after GPCR activation [40]. Recently a role for G_i/o_ βγ subunits as modulators of Gα_q_ activation of PLC, forming a Gα_q_-PLC-G_i/o_ βγ complex, and depending on the affinity for the plasma membrane of the γ subunits has been proposed [41]. Regarding potential interactions between CaSR- and cannabinoid-dependent pathways modulating [Ca^2+^]_i_, it is conceivable that WIN55212–2-dependent CB_1_R activation increases [Ca^2+^]_i_ by augmenting CaSR-Gα_q_-dependent activation of PLC with the participation of CB_1_R-mediated G_i/o_ βγ release in neuroblastoma cells.

Two conundrums remain. One is that only the WIN55212-2 but no other cannabinoid or endocannabinoid agonist family representatives could stimulate the Ca^2+^ mobilization. The other is that we observed a role for CaSR in WIN55212-2/CB_1_R-dependent increases in [Ca^2+^]_i_ at low eCa^2+^ and not at high eCa^2+^. These results suggest the possibility of different Gα_q_-PLC-G_i/o_ βγ complexes being formed depending on eCa^2+^ and the cellular proximity of the receptors and the G proteins to which they are pre-coupled (Figure 6). For example, our earlier studies demonstrated that WIN55212-2 behaves as an agonist for all three Gi subtypes, whereas the THC analog desacetyllevonantradol behaves as an agonist for Gi_1_ and Gi_2_ but an inverse agonist for Gi_3_; and methanandamide behaves as an agonist at Gi_3_ but an inverse agonist for Gi_1_ and Gi_2_ [42].

In the current study, we also showed a role for nonCB_1_/CB_2_ receptors in controlling [Ca^2+^]_i_. Among the candidates for these receptors is GPR55, which can be activated by CB_1_R antagonist/inverse agonist AM251 (but not SR141716), and the lysophospholipid, lysophosphatidylinositol (LPI). GPR55 utilizes Gα_q_ or Gα_12/13_ for signal transduction [43] and can promote Ca^2+^ mobilization and mitogen-activated protein kinase (MAPK) phosphorylation [44]. GPR55 modulates neurotransmitter release through modulation of neuronal [Ca^2+^]_i_ [45]. Activation of GPR55 in dorsal root ganglia neurons by various cannabinoids, including Δ^9^-THC and methanandamide, increases [Ca^2+^]_i_ through a mechanism involving Gα_q_, PLC, and IP_3_ receptors [7].

Our results with O-1918, could implicate a role for GPR55. However, since WIN55212-2 fails to activate [Ca^2+^]_i_ by GPR55 [7] the effects of O-1918 on WIN55212-2-dependent responses we observed may be explained by the CB_1_-induced activation of phospholipase A and synthesis of LPI, which in turn would activate GPR55. This possibility has been suggested to explain the observed actions of CB_1_R antagonists/inverse agonists on GPR55-mediated actions [46].

Another possibility for the nonCB_1_/CB_2_ receptor is GPR18, which originally was characterized by its activation by abnormal cannabidiol (abn-CBD) and inhibition by CBD and O-1918, but now de-orphanized as a GPCR activated by the endogenous anandamide metabolite, N-arachidonoyl-glycine (see review for original references [34,37]). In exogenous expression models, GPR18 responds to N-arachidoyl-glycine by Ca^2+^ mobilization [47]. GPR18 also responds to the inflammation pro-resolving polyunsaturated, hydroxylated 22-C lipid, resolvin D2 (RvD2). In monocytes, macrophages, microglia, and BV2 microglia, and polymorphonuclear neutrophils, GPR18 couples to G_i/o_, G_q/11_, and G_s_ to signal by increasing cAMP and protein kinase A (PKA), and phosphorylation of signal transducer and activator of transcription 3 (STAT3) (for review and original references, see [37,48]). However, WIN55212-2 has not been demonstrated to elicit any signaling responses by GPR18 (see summary tables and text for original references [37,48]).

A final possibility is that WIN55212-2 might activate a putative Alkyl Indole receptor as described for microglia and astrocyte cellular signaling in response to WIN55212-2 and analogs [49,50]. Such receptors may be those identified as [^3^H]WIN55212-2 binding sites in neuroblastoma-glioma hybrid NG108-15 cells [51]. The potential for a specific WIN55212-2 “receptor” in brain membranes was suggested by studies of the C57Bl/6 CB_1_ knock-out mouse which showed that anandamide and WIN55212-2 could promote G protein activation ([^35^S]GTPγS binding) [52]. The properties of a putative Alkyl Indole receptor have yet to be fully characterized.

In summary, the aminoalkylindole WIN55212-2 can modulate [Ca^2+^]_i_ in N18TG2 cells via CB_1_R-dependent and independent mechanisms. We observed functional interactions between the CB_1_R and CaSR activation in regulating [Ca^2+^]_i_, and these interactions are dependent on eCa^2+^ with the participation of nonCB_1_/CB_2_ receptors in neuroblastoma cells. Future studies should address the effects of LPI, abn-CBD, CBD and RvD2 to determine the role of GPR55 and GPR18 on Ca^2+^ mobilization influenced by the CaSR. Additionally, studies should address whether aminoalkylindole analogs that act on the putative WIN55212-2 Alkyl Indole “receptor” are involved in the regulation of [Ca^2+^]_i_ by CaSR. Details regarding the mechanism by which the CaSR interfaces with Class A GPCRs to regulate intracellular calcium stores could be analyzed using thapsigargin-dependent control of [Ca^2+^]_i_ [6] and other current methods of structural and functional analysis.

## Figures and Tables

**Figure 1 cells-11-02947-f001:**
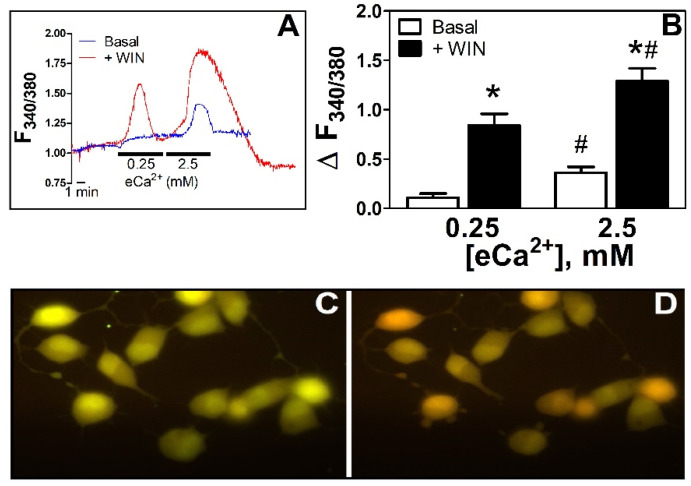
**WIN55212-2 increases [Ca^2+^]_i_: effect of eCa^2+^**. (**A**). Time course of the changes in F_340/380_ in N18TG2 cells incubated in 0.25 and 2.5 mM eCa^2+^ in basal conditions (blue line) or the presence of WIN55212-2 5 µM (red line). (**B**). Relative changes in [Ca^2+^]_i_ as ΔF_340/380_ in basal conditions (Basal, □, *n* = 6) or in the presence of WIN55212-2 5 µM (+WIN, ■, *n* = 7). * *p* < 0.05 vs. basal; # *p* < 0.05 vs. 0.25 mM eCa^2+^. (**C**). Fluorescence image from a representative coverslip with N18TG2 cells observed under the imaging system used (see section Imaging in Materials and Methods) in eCa^2+^ 0.25 mM. 40× amplification. (**D**). Same cells as in C after 2 min treatment (approximately at peak response) with WIN55212-2 5 µM in eCa^2+^ 0.25 mM, 40x amplification. Changes in pseudo color from green to red represent the increase in emission after excitation at 340 nm and decrease in emission after excitation at 380 nm of Fura-2 upon binding to Ca^2+^, the basis of the ratiometric (F_340/380_) system for determinations of relative changes in [Ca^2+^]_i_ [17].

**Figure 2 cells-11-02947-f002:**
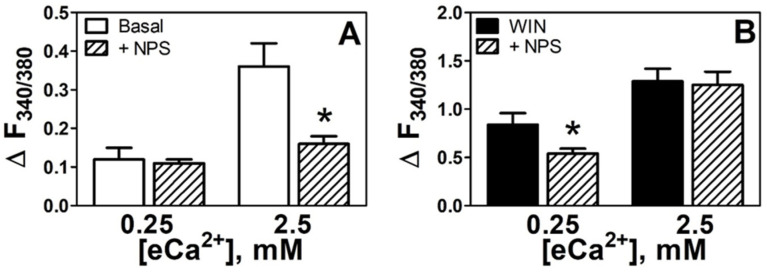
**Increases in [Ca^2+^]_i_ depend on the CaSR.** (**A**). Effects of the CaSR inhibitor NPS2314 3 µM (+NPS, diagonal stripes upward bars, *n* = 7) on [Ca^2+^]_i_ in basal conditions (open bars, *n* = 6). (**B**). Effects of the CaSR inhibitor NPS2314 3 µM (+NPS, diagonal stripes upward bars, *n* = 7) on [Ca^2+^]_i_ in the presence of WIN55212-2 5 µM (black bars, *n* = 7). * *p* < 0.05 vs. basal or WIN55212-2 at the corresponding eCa^2+^.

**Figure 3 cells-11-02947-f003:**
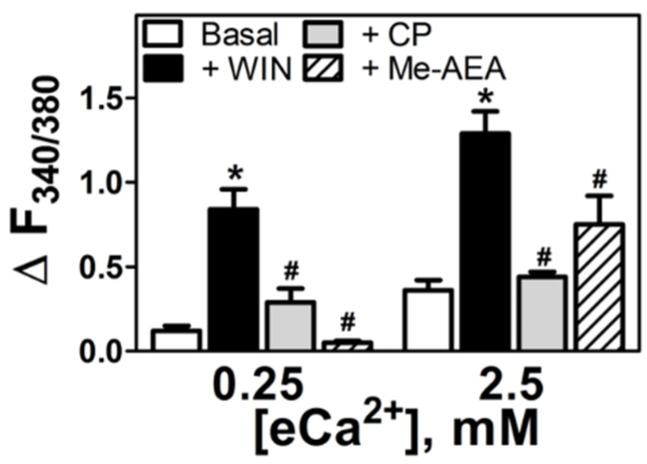
**Aminoalkylindole-specific increase in [Ca^2+^]_i_ in N18TG2 cells.** Increases in [Ca^2+^]_i_ in conditions of low (0.25 mM) and high (2.5 mM) eCa^2+^ in the presence of WIN55212-2 5µM (+WIN, *n* = 7), the bicyclic mimetic of THC CP55940 5 µM (+CP, *n* = 4), or the arachidonoylethanolamine analog meth-anandamide 5 µM (+Me-AEA, *n* = 5) * *p* < 0.05 vs. basal; # *p* < 0.05 vs. WIN55212-2.

**Figure 4 cells-11-02947-f004:**
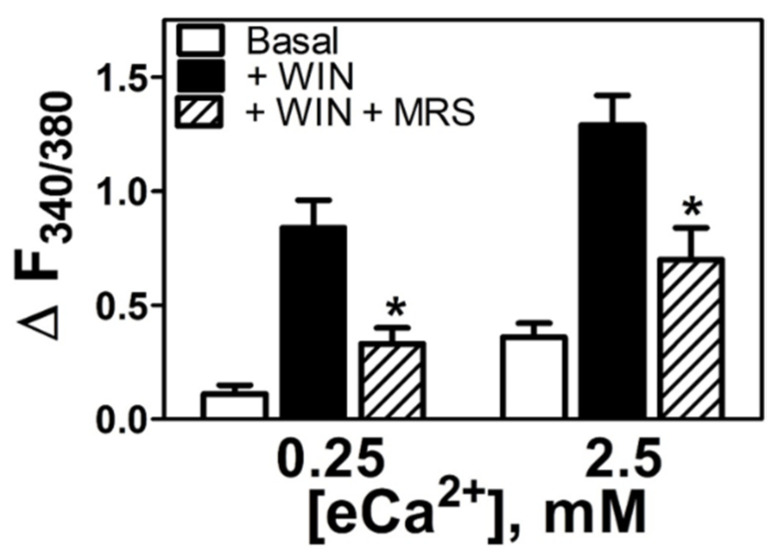
**WIN55212-2-mediated increase in [Ca^2+^]_i_ is dependent on store operated calcium entry.** Increases in [Ca^2+^]_i_ in low (0.25 mM) and high (2.5 mM) eCa^2+^ in basal conditions (Basal, *n* = 6), in the presence of WIN55212-2 (5 µM, *n* = 7) (+WIN), and in the presence of WIN55212-2 plus MRS1845 (+WIN + MRS, *n* = 4 or *n* = 7) (10 µM). * *p* < 0.05 vs. +WIN.

**Figure 5 cells-11-02947-f005:**
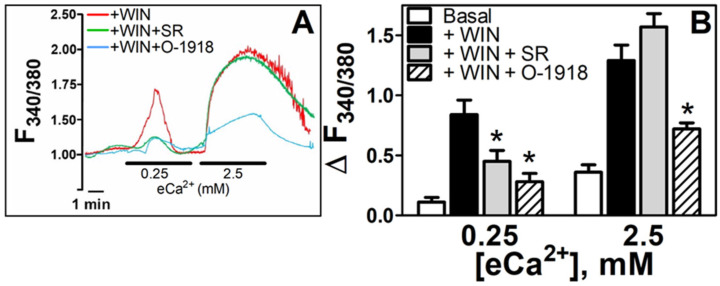
**WIN55212-2-mediated increase in [Ca^2+^]_i_ is dependent on the CB_1_ receptor and on nonCB_1_/CB_2_ receptors.** (**A**). Time course of the changes in F_340/380_ in N18TG2 cells incubated in 0.25 and 2.5 mM eCa^2+^ in the presence of WIN55212-2 5 µM (+WIN, red line), SR141716 1 µM (+WIN + SR, green line) and WIN55212-2 plus O-1918 10 µM (+WIN + O-1918, light blue line). (**B**). Increases in [Ca^2+^]_i_ in low (0.25 mM) and high (2.5 mM) eCa^2+^ in basal conditions (Basal, open bars, *n* = 6), in the presence of WIN55212-2 (5 µM) (+WIN, *n* = 7), in the presence of WIN55212-2 plus SR141716 (1 µM) (+WIN + SR, *n* = 7), and in the presence of WIN55212-2 plus O-1918 (10 µM) (+WIN + O-1918, *n* = 5). * *p* < 0.05 vs. +WIN.

**Figure 6 cells-11-02947-f006:**
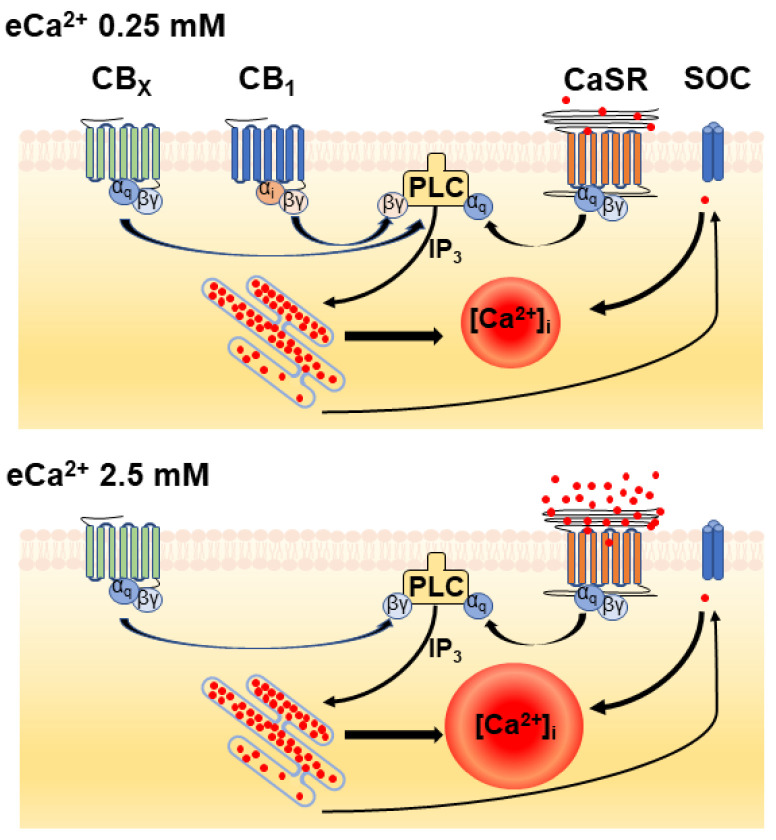
**Diagram illustrating potential functional interactions between cannabinoid receptors and CaSR in neuroblastoma cells in different levels of eCa^2+^.** The formation of different Gα_q_-PLC-G_i/o_ βγ complexes depending on eCa^2+^ and cellular proximity of the GPCRs involved is proposed. CB_1_: CB_1_ receptor. CBx: putative ‘WIN55212-2’ receptor. CaSR: calcium sensing receptor. SOC: store operated calcium channels.

## Data Availability

Not applicable.

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
