# Peer review of "WIN55212-2 Modulates Intracellular Calcium via CB1 Receptor-Dependent and Independent Mechanisms in Neuroblastoma Cells"

_cells, 2022, doi:10.3390/cells11192947_

Round 1

Reviewer 1 Report

The submitted research article analyses the effects of WIN55212-2 on the Intracellular Calcium in Neuroblastoma Cells; CB1 receptor dependent and independent mechanisms have been reported. The manuscript is well done and informative, the experimental design is appropriate with agonists/antagonists used alone or in combination; presented data sustain the conclision of the study.

The suggested minor changes aim at further improve the manuscript.

·         The specific agonists/antagonists have to be better and clearly presented in the main text

·         A rationale for the selected drug doses is missing (i.e.add references to previous studies if available)

·         Administration/treatments with drug have to be better explained in the method section

·         There is not full correspondence between Figure 2b, and figure 2 legends (i.e. missing +WIN and + WIN + NPS

·         Abbreviations have not been always defyned at the first appearence in the main text

·         Uniformate the format of figure legends

·         Reference list does not seems formatted accordingly to MDPI style

Author Response

Answers to Reviewer 1: please see answers to Reviewer 1 comments in red font below.

The submitted research article analyses the effects of WIN55212-2 on the Intracellular Calcium in Neuroblastoma Cells; CB1 receptor dependent and independent mechanisms have been reported. The manuscript is well done and informative, the experimental design is appropriate with agonists/antagonists used alone or in combination; presented data sustain the conclision of the study.

We thank the reviewer for the encouraging comments.

The suggested minor changes aim at further improve the manuscript.

  • The specific agonists/antagonists have to be better and clearly presented in the main text

The definitions of agonists and antagonists used in the experiments is now presented in the Results section of the manuscript.

  • A rationale for the selected drug doses is missing (i.e.add references to previous studies if available)

Doses of drugs used were based on the literature; the Results section now includes appropriate references.

  • Administration/treatments with drug have to be better explained in the method section

The subsection ‘Imaging’ in ‘Materials and Methods’ was expanded to include a detailed description of the treatments and drug administration protocol. The section ‘Aminoalkylindole-specific potentiation of the eCa2+-mediated increase in [Ca2+]i’ provided a rationale for using different CBR agonists.

  • There is not full correspondence between Figure 2b, and figure 2 legends (i.e. missing +WIN and + WIN + NPS

We have edited Figure Legends for consistency. Figures with two panels contain clear subdivisions A and B, such as Figure 2 and the now modified Figure 5. In addition, two extra panels were added to Figure 1, and the legend was edited accordingly. Legends were also edited to reflect the graphs’ bars labels.

  • Abbreviations have not been always defyned at the first appearence in the main text

Abbreviations are now defined at their first appearance in the manuscript.

  • Uniformate the format of figure legends

Figure legends were reformatted for consistency, description of the pattern in each bar was included

  • Reference list does not seems formatted accordingly to MDPI style

The reference list was updated and formatted following the MDPI style

Reviewer 2 Report

Comment:

This paper discusses "WIN55212-2 modulates Intracellular Calcium via CB1 receptor dependent and independent Mechanisms in Neuroblastoma Cells. ". The main contribution of the paper is "They investigated the mechanisms involved in the ([Ca2+ 18 ]i increase in N18TG2 neuroblastoma cells, which endogenously express both receptors."

This is an interesting study and is generally well written and structured. However, in my opinion the paper has some shortcomings in regards to signaling.

In several instances I also strongly suggest to cite more relevant and recent literature.

Important papers

Cannabinoids

1. The Impact of CB1 Receptor on Inflammation in Skeletal Muscle Cells

https://pubmed.ncbi.nlm.nih.gov/34421307/

2. The Impact of CB1 Receptor on Nuclear Receptors in Skeletal Muscle Cells

https://pubmed.ncbi.nlm.nih.gov/35366244/

Minor comments:

·       Well written except in some situations. I advise recheck it again.

·       The introduction should be advised to be re-written to be in more logical flow.

·       Cite more recent papers in introduction since there is no references in some parts. (above is suggested)

·       Please, suggest future experiments in details.

·       Please, Specify the most specific protein that you suggest might be related lo signalling.

·       Although it needs to be in more logical flow, the introduction provides a good, generalized background of the topic. However, why not cite more literature papers (above).

·       I think the motivations for this study need to be made clearer.

·       Regarding the figures: I recommend make more figures to be illustrative.

Given these shortcomings the manuscript requires Minor revisions.

"I recommend that this paper be accepted after minor revision."

Author Response

Cells_1892772_R1

Answers to Reviewer 2: please see answers to Reviewer 2 comments in red font below.

This paper discusses "WIN55212-2 modulates Intracellular Calcium via CB1 receptor dependent and independent Mechanisms in Neuroblastoma Cells. ". The main contribution of the paper is "They investigated the mechanisms involved in the ([Ca2+ 18 ]i increase in N18TG2 neuroblastoma cells, which endogenously express both receptors."

This is an interesting study and is generally well written and structured. However, in my opinion the paper has some shortcomings in regards to signaling.

In several instances I also strongly suggest to cite more relevant and recent literature.

Important papers

Cannabinoids

  1. The Impact of CB1 Receptor on Inflammation in Skeletal Muscle Cells

https://pubmed.ncbi.nlm.nih.gov/34421307/

  1. The Impact of CB1 Receptor on Nuclear Receptors in Skeletal Muscle Cells

https://pubmed.ncbi.nlm.nih.gov/35366244/

Minor comments:

  • Well written except in some situations. I advise recheck it again.
  • The introduction should be advised to be re-written to be in more logical flow.

The introduction was edited to provide a better flow

  • Cite more recent papers in introduction since there is no references in some parts. (above is suggested)

Seven new references were added, including one suggested by the reviewer.

  • Please, suggest future experiments in details.

Future experiments are now presented in the Discussion

  • Please, Specify the most specific protein that you suggest might be related lo signalling.

As presented in the Discussion, our results suggest the possibility of different Gαq-PLC-Gi/o βγ complexes being formed depending on eCa2+.  Thus, the cellular proximity of the receptors and the G proteins to which they are pre-coupled will determine the activation of PLC and the transduction of intracellular signals. Additionally, this central role of PLC is presented in Figure 6.

  • Although it needs to be in more logical flow, the introduction provides a good, generalized background of the topic. However, why not cite more literature papers (above).

One of the papers suggested by the reviewer is now cited

  • I think the motivations for this study need to be made clearer.

A paragraph was inserted in the Introduction to expand and clarify the motivation for this study.

  • Regarding the figures: I recommend make more figures to be illustrative.

For the purposes of the results included in the manuscript we estimate that the figures included present a clear picture of the data presented. Following reviewer’s recommendation new figure panels were added to Figure 1 and to Figure 5. Figure 6 was also edited.

Given these shortcomings the manuscript requires Minor revisions.

"I recommend that this paper be accepted after minor revision."

Thanks.

Round 2

Reviewer 1 Report

The authors carried out all the suggested changes; I endorse the manuscript for publication.